# Automatic classification of lymphoma lesions in FDG-PET–Differentiation between tumor and non-tumor uptake

**Thomas W Georgi**[1]*, **Axel Zieschank**[2], **Kevin Kornrumpf**[2], **Lars Kurch**[1], **Osama Sabri**[1], **Dieter Körholz**[3], **Christine Mauz-Körholz**[3], **Regine Kluge**[1‡], **Stefan Posch**[2‡]

**1** Department of Nuclear Medicine, University of Leipzig, Leipzig, Germany, **2** Institute of Computer Science, Martin-Luther-University of Halle and Wittenberg, Halle, Saale, Germany, **3** Department of Pediatric Oncology, Justus-Liebig-University, Giessen, Germany

☯ These authors contributed equally to this work.
‡ These authors also contributed equally to this work.
* Thomas.Georgi@medizin.uni-leipzig.de

## Abstract

### Introduction

The automatic classification of lymphoma lesions in PET is a main topic of ongoing research. An automatic algorithm would enable the swift evaluation of PET parameters, like texture and heterogeneity markers, concerning their prognostic value for patients outcome in large datasets. Moreover, the determination of the metabolic tumor volume would be facilitated. The aim of our study was the development and evaluation of an automatic algorithm for segmentation and classification of lymphoma lesions in PET.

### Methods

Pre-treatment PET scans from 60 Hodgkin lymphoma patients from the EuroNet-PHL-C1 trial were evaluated. A watershed algorithm was used for segmentation. For standardization of the scan length, an automatic cropping algorithm was developed. All segmented volumes were manually classified into one of 14 categories. The random forest method and a nested cross-validation was used for automatic classification and evaluation.

### Results

Overall, 853 volumes were segmented and classified. 203/246 tumor lesions and 554/607 non-tumor volumes were classified correctly by the automatic algorithm, corresponding to a sensitivity, a specificity, a positive and a negative predictive value of 83%, 91%, 79% and 93%. In 44/60 (73%) patients, all tumor lesions were correctly classified. In ten out of the 16 patients with misclassified tumor lesions, only one false-negative tumor lesion occurred. The automatic classification of focal gastrointestinal uptake, brown fat tissue and composed volumes consisting of more than one tissue was challenging.

**Data Availability Statement:** All relevant data are within the manuscript and its supporting information files.

**Funding:** The study was supported by the foundation "Mitteldeutsche Kinderkrebsforschung" ("Children's cancer research of central germany") https://www.kinderkrebsforschung.net The funders had no role in study design, data collection and analysis, decision to publish, or preparation of the manuscript.

**Competing interests:** The authors have declared that no competing interests exist.

## Conclusion

Our algorithm, trained on a small number of patients and on PET information only, showed a good performance and is suitable for automatic lymphoma classification.

## Introduction

Fluorodeoxyglucose positron emission tomography (PET) is a well-established, sensitive method for the assessment of Hodgkin lymphoma (HL) and glucose-avid Non-HL [1–4]. The initial PET scan is important for exact staging and radiotherapy planning [5, 6].

Lymphoma represent a systemic disease in which multifocal involvement pattern is common [7, 8]. The precise detection of all initial lymphoma lesions is crucial for correct treatment stratification [9, 10]. The differentiation between lymphoma uptake and non-tumor uptake in PET can be challenging [11]. Reasons are the multifocal lymphoma involvement pattern with large variability in terms of size, localization and tracer uptake, as well as the complex physiological FDG distribution with variable uptake behavior, e.g. of the heart, the kidneys and the skeleton. In addition, inflammatory foci, bone marrow activation and activated brown fat tissue may hamper the PET evaluation. The assessment of PET scans of lymphoma patients is feasible for experienced nuclear physicians but quite challenging for automatic algorithms.

The initial metabolic tumor volume was reported to be an independent prognostic factor in lymphoma patients [12]. The prognostic value of further PET parameters, like texture and heterogeneity markers is subject of current research [13–17]. Large numbers of PET datasets are required to evaluate the prognostic value of different parameters. Manual detection and segmentation of lymphoma lesions is very time consuming and therefore not suitable for large datasets. Algorithms have to be developed and tested, enabling the automatic segmentation of lymphoma lesions. Moreover, an automatic classification of non-tumor lesions in different tissue categories could support dosimetry calculations.

The aim of our study was to develop and evaluate an automatic algorithm for segmentation and classification of lymphoma lesions in PET. In addition, we aimed at an optimized differentiation of the various non-tumor tissues.

## Methods

### Patients

Our study included pre-treatment PET scans from HL patients from the EuroNet-PHL-C1 (C1) trial (EudraCT number NCT00433459) [18]. PET imaging was performed in multiple PET centers, following the imaging recommendations of the C1 study protocol.

Ethical approval was granted by the ethics committee of the University of Leipzig (132/19-ek). C1 study patients and/or their guardians gave written informed consent and acknowledged transfer and storage of their imaging data on the Pediatric-Hodgkin-Network server [19]. For our evaluation, all PET scans were completely anonymized. Thus, the ethics committee waived the requirement for additional informed consent.

### Data preparation and transfer

In preparation of an evaluation regarding prognostic PET parameters for relapse, each initial PET scan from the C1 study was anonymized and labeled with two attributes only. The first attribute represented the PET result of the patient after two cycles of chemotherapy ("PET-

positive" or "PET-negative"). The second attribute indicated whether a relapse occurred later on or not ("relapse" or "non-relapse"). Thus, each anonymized PET scan could be assigned to one of four subgroups (PET-positive relapse, PET-positive non-relapse, PET-negative relapse, PET-negative non-relapse). All anonymized PET scans were transferred via secured data connection from the Pediatric-Hodgkin-Network server to the Institute of Computer Science at the Martin-Luther-University of Halle (Saale), Germany for further evaluation.

### Preprocessing

For all PET scans, the intensity values were smoothed for denoising using a Gauss filter with σ = 2.0. Thereafter, SUV values were calculated and coronal SUV projections were reconstructed for visual plausibility check. From the valid scans, 15 PET scans from each subgroup, a total of 60 scans, were drawn randomly for this study.

### Segmentation

First, all voxels with an SUV below 2.5 were excluded. Second, a watershed algorithm was used for segmentation of the SUV dataset [20]. This algorithm uses the concept that individual voxels belong to certain catchment basins to define different volumes. Watersheds represent borders between these basins. To detect watersheds, the flooding of the topography is simulated. The algorithm stops when all voxels are assigned to either one catchment area or a watershed. Third, neighboring volumes were merged based on similar texture. Specifically, for each volume co-occurrence matrices were computed using a distance of one for all directions and the average of the resulting contrast, entropy and inverse difference moment was determined. Two volumes were merged if the absolute differences of their aggregated texture measures was smaller than a threshold of two.

Volumes smaller than 2.0 ml were excluded from further analysis [12, 21].

### Automatic cropping algorithm

For further standardization, each PET scan was cropped along the z-axis. An heuristic algorithm was developed, cropping the PET scan at brain and urinary bladder level. An illustrated description is given in the supplementary material. Segmented volumes were not considered for further analysis if they overlapped the boundaries or laid outside.

### Manual classification

All segmented volumes were manually classified by a nuclear physician with ten years of expertise in HL evaluation. Each volume was assigned to one of the following 14 categories: tumor, composed volumes of tumor and non-tumor tissue (T+NT), brain, skeleton, head-and-neck-region, heart, right kidney, left kidney, liver, gastrointestinal tract, genital organs, urinary bladder, activated brown fat tissue and composed volumes consisting of more than one non-tumor tissue (NT+NT).

### Feature computation and normalization

In our study, 31 features were used for classification: 19 SUV-based features, six shape-based features and six spatial location features (Table 1). These features had to be normalized to be comparable between scans, since patients differed in their constitutions and PET scanner in their resolutions. Our approach was to compute the features in the original PET scan and subsequently transform them to a standardized size and scanner resolution.

**Table 1. Features of the segmented volumes used for classification.**

| Features for Classification | | | |
|---|---|---|---|
| SUV-based features | | Shape-based features | Spatial location features |
| Maximum SUV | Mean absolute deviation of SUV | Area surface ratio | Direction of major axis (x) |
| Minimum SUV | Root mean squared of SUV | Surface difference[2] | Direction of major axis (y) |
| Mean SUV | Standard deviation of SUV | Scaled surface area | Direction of major axis (z) |
| Median SUV | Skewness of SUV | Scaled volume | Location of centorid (x) |
| Range of SUV | Coarseness of NGTDM | Compactness | Location of centorid (y) |
| Variance of SUV | Contrast of NGTDM | Maximum diameter | Location of centorid (z) |
| Energy of SUV | Busyness of NGTDM | | |
| Entropy of SUV | Complexity of NGTDM | | |
| Kurtosis of SUV | Outside difference[1] | | |
| Uniformity of SUV | | | |

1) Mean difference of SUV values between volume and surrounding.

2) Estimated average ratio of eucliden vs surface distance of surface voxel pairs.

Abbreviations: SUV—Standard uptake value, NGTDM—Neighboring gray tone difference matrix.

## Automatic classification

For automatic classification of the segmented volumes, the random forest method [22] was applied. A random forest consists of several hundred up to thousands of decision trees. Each tree consists of a hierarchy of binary decisions. With every decision it was checked, if a specific feature of the volume exceeded a cut-off value or not (e.g. Is the maximum SUV above 4.0?). After navigating through the whole tree up to the leaf, the given volume was classified into the category associated with this leaf. Each volume was classified using all decision trees. The final classification result represented the majority vote of all trees.

A nested cross validation was used for training of the random forest. For ease of presentation, only an overview is given here, a detailed description can be found in the supplementary material. The whole dataset was split into three parts: a training set, a validation set and a test set. Using the training set, the random forest was trained. All decision trees of the forest were trained independently. The validation set was used to find the best combination of hyperparameters. Hyperparameters in our study were: the maximum depth of each decision tree and the number of selectable features for each decision. The trained random forest with the best hyperparameter combination was applied to the test set. This procedure was repeated until each part of the entire data set was once a test data set to derive an unbiased estimate of the performance.

## Performance evaluation

The results of the automatic cropping algorithm were evaluated visually. Nested cross validation with random data partitioning and ten repetitions was used for performance evaluation of the automatic classification. The classification results were averaged across the ten repetitions and subsequently rounded to integral numbers. For the binary decision between tumor and non-tumor, T+NT volumes were considered tumor lesions since our aim was to include all tumor lesions. Sensitivity, specificity, positive predictive value (PPV), negative predictive value (NPV) and F1-score (F1 = 2 / (sensitivity$^{-1}$ + PPV$^{-1}$)) were calculated.

A flowchart of the methodical workflow is given in Fig 1.

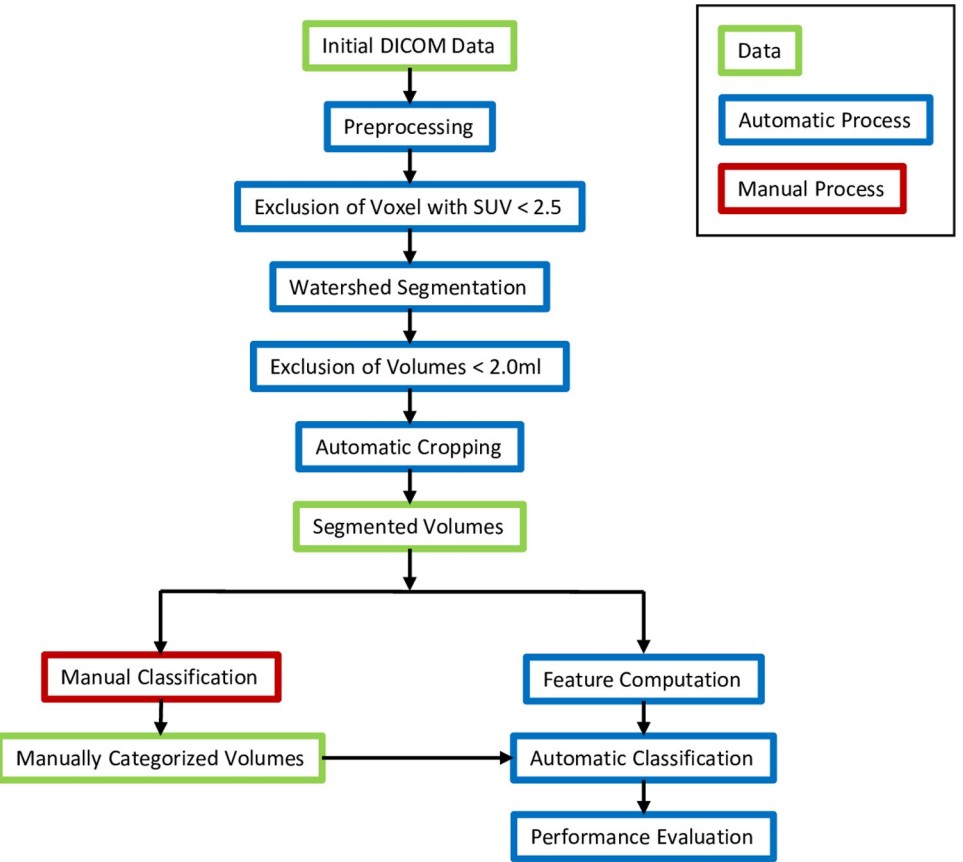

**Fig 1. Methodical workflow of the classification algorithm.**

## Results

### Evaluation of the automatic cropping algorithm

The automatic cropping at brain level was successful in 50/60 patients. In nine out of the ten patients with incorrect cropping, the brain was not included in the PET acquisition. As a result, the scan was cropped at the salivary glands in six patients and at a tumor lesion in the neck in three patients. In one patient, the PET scan was cropped at a tumor lesion in the neck although the brain was visible. Due to incorrect cropping in the upper part of the scan, five tumor lesions in the neck were excluded from further evaluation.

The automatic cropping at bladder level was successful in 57/60 patients. In one patient, the scan was cropped above the bladder at the iliac crest, in one patient below the bladder at the testicles and in one patient no lower boundary was defined. However no tumor lesion got lost on the basis of incorrect cropping at the lower part of the scan. In two patients, tumor lesions were excluded from further evaluation since they were located below bladder level—inguinal lymph nodes in one patient and one skeletal lesion in the right femur in another patient.

### Segmentation

Overall, 853 segmented volumes were equal or larger than 2.0 ml and located completely within the cropping boundaries and therefore considered for further evaluation.

## Manual classification

225 tumor lesions and 21 composed T+NT volumes were manually classified, an average of 4.1 tumor lesions per patient. Tumor lesions occurred in lymph nodes above and below the diaphragm in 59 and 17 patients, respectively. Extra-nodal lesions occurred in the lung, the skeleton, the liver and the spleen in eight, five, one and ten patients, respectively.

The other 607 volumes were non-tumor tissues, including six composed NT+NT volumes. Most frequently, physiological skeletal uptake was segmented (n = 295), mainly located in spine and pelvis. Tracer uptake in the kidneys was also frequently segmented (n = 134), occasionally in multiple volumes per kidney due to uneven tracer distribution. 86 non-tumor volumes were located in the head-and-neck-region, mostly in the salivary glands, the Waldeyer´s ring or the vocal cord. Due to incorrect cropping, two urinary bladders were included in the further evaluation. An overview of all manually classified volumes is shown in the last line of Table 2.

## Automatic classification of tumor vs. non-tumor

Overall, 203/246 tumor lesions and 554/607 non-tumor volumes were classified correctly, corresponding to a sensitivity of 83%, a specificity of 91%, a PPV of 79%, a NPV of 93% and an f1-score of 81%.

43 tumor lesions were misclassified. These lesions were mistakenly considered to be non-tumor uptake, mainly in the skeleton (n = 17), the head-and-neck-region (n = 12) or the kidney (n = 9). 53 non-tumor volumes were misclassified as tumor lesions. Most of these misclassified lesions referred to non-malignant uptake in the gastrointestinal tract (n = 13), physiological uptake in the skeleton (n = 9) or uptake in activated brown fat tissue (n = 7) (Table 2).

## Automatic classification of non-tumor tissues

The most common non-tumor tissues (skeleton, kidneys and soft tissue in the head-and-neck-region) were correctly classified in 94%, 90% and 94% of all cases. 77% of the heart volumes

**Table 2. Confusion matrix of manual and automatic classification of the segmented volumes.**

| | | Manual Classification | | | | | | | | | | | | |
|---|---|---|---|---|---|---|---|---|---|---|---|---|---|---|
| | | T | T+NT | SK | HN | HT | RK | LK | LI | GI | G | BL | BF | NT+NT | sum |
| Automatic Classification | T | 182 | 7 | 9 | 5 | 5 | 1 | 5 | 0 | 13 | 2 | 1 | 7 | 3 | 240 |
| | T+NT | 4 | 10 | 0 | 0 | 0 | 0 | 0 | 0 | 0 | 0 | 0 | 0 | 2 | 16 |
| | SK | 15 | 2 | 277 | 0 | 2 | 4 | 0 | 1 | 6 | 0 | 1 | 1 | 0 | 309 |
| | HN | 11 | 1 | 4 | 81 | 0 | 0 | 0 | 0 | 0 | 0 | 0 | 1 | 0 | 98 |
| | HAT | 1 | 0 | 0 | 0 | 24 | 0 | 0 | 0 | 0 | 0 | 0 | 0 | 0 | 25 |
| | RK | 2 | 0 | 1 | 0 | 0 | 61 | 0 | 0 | 0 | 0 | 0 | 0 | 0 | 64 |
| | LK | 6 | 1 | 1 | 0 | 0 | 0 | 60 | 0 | 0 | 0 | 0 | 1 | 0 | 69 |
| | LI | 0 | 0 | 0 | 0 | 0 | 0 | 0 | 0 | 0 | 0 | 0 | 0 | 0 | 0 |
| | GI | 3 | 0 | 3 | 0 | 0 | 1 | 2 | 0 | 17 | 1 | 0 | 0 | 1 | 28 |
| | G | 0 | 0 | 0 | 0 | 0 | 0 | 0 | 0 | 1 | 0 | 0 | 0 | 0 | 1 |
| | BL | 0 | 0 | 0 | 0 | 0 | 0 | 0 | 0 | 0 | 0 | 0 | 0 | 0 | 0 |
| | BF | 1 | 0 | 0 | 0 | 0 | 0 | 0 | 0 | 0 | 0 | 0 | 2 | 0 | 3 |
| | NT+NT | 0 | 0 | 0 | 0 | 0 | 0 | 0 | 0 | 0 | 0 | 0 | 0 | 0 | 0 |
| | sum | 225 | 21 | 295 | 86 | 31 | 67 | 67 | 1 | 37 | 3 | 2 | 12 | 6 | 853 |

Abbreviations: T—Tumor, T+NT—Composed volume of tumor and non-tumor tissue, SK—Skeleton, HN—Head-and-neck-region, HT—Heart, RT- Right kidney, LK —Left kidney, LI—Liver, GI—Gastrointestinal tract, G—Genital organs, BL—Urinary bladder, BF—Activated brown fat tissue, NT+NT—composed volumes of more than one non-tumor tissue type (NT+NT).

were correctly classified. The automatic classification of gastrointestinal uptake, activated brown fat tissue and composed NT+NT volumes was an issue. The rates for correct classification were 46% (17/37), 17% (2/12) and 0% (0/6), respectively.

## Results on patient level

In 44/60 (73%) patients, all tumor lesions were correctly classified. In ten out of the 16 patients with misclassified tumor lesions, only one false-negative tumor lesion occurred. Misclassified tumor lesions were located in all body regions: in six patients in the neck, in four patients in the mediastinum and in six patients below the diaphragm. The algorithm missed splenic involvement in one patient and skeletal involvement in five patients.

Non-tumor uptake misclassified as tumor was seen in 31/60 (52%) patients, mainly below the diaphragm (n = 17). In 14/31 patients, only one non-tumor lesion was misclassified. In seven patients, physiological skeletal uptake was considered to be tumor.

In 13/37 patients with cardiac uptake, the heart was part of a composed volume. In 18 out of the other 24 patients (75%), the heart was correctly classified. In 3/6 patients with cardiac misclassification, the heart was untypically configured since only the valve plane showed increased tracer uptake Overall, in 22 patients all volumes were correctly classified. In nine patients both, misclassified tumor and non-tumor lesions were seen. An example for each type is shown in Figs 2 and 3.

## Discussion

Our automatic algorithm for segmentation and classification of lymphoma lesions in initial PET scans resulted in a sensitivity of 83% and a specificity of 91%. In 44 of our 60 patients, all tumor lesions were classified correctly. In ten patients only one tumor lesion was misclassified and in six patients more than one. Most frequently, tumor lesions were misclassified as non-malignant uptake in the skeleton or the head-and-neck-region.

Tumor lesions were discriminated from eleven different non-tumor uptake categories, including complex areas like salivary glands, skeleton and brown fat tissue. The classification of non-tumor uptake provided sensitivities ≥ 90% for the most common tissue types (skeleton, kidneys and soft tissue in the head-and-neck-region). These three tissues together amount to 85% of the number of all non-tumor volumes.

The automatic heart classification is challenging [23] due to the variable uptake behavior. In half of our patients with misclassified heart volumes an untypical cardiac uptake pattern was seen. However, 77% of all heart volumes were classified correctly.

The classification of non-malignant gastrointestinal uptake was difficult. Due to its focal appearance and variable localization, gastrointestinal uptake may appear like lymphoma lesions. In our study, more than half of all gastrointestinal volumes were misclassified, one third was mistakenly classified as tumor lesion.

Another critical category for automatic classification was activated brown fat tissue. Typically, it is characterized by multiple singular uptake areas mostly located in the neck and shoulder region. This area is also a preferred localization for lymphoma lesions. Thus, increased uptake in the brown fat tissue hampers the detection of lymphoma uptake, for physicians but especially for automatic algorithms.

Overall, in about half of our patients, at least one non-tumor volume was misclassified as tumor lesion.

For our algorithm the following conditions were chosen:

First, an SUV threshold of 2.5 for lesion segmentation was applied. This sensitive threshold [24, 25] was chosen to include all potential tumor lesions in our evaluation. Different

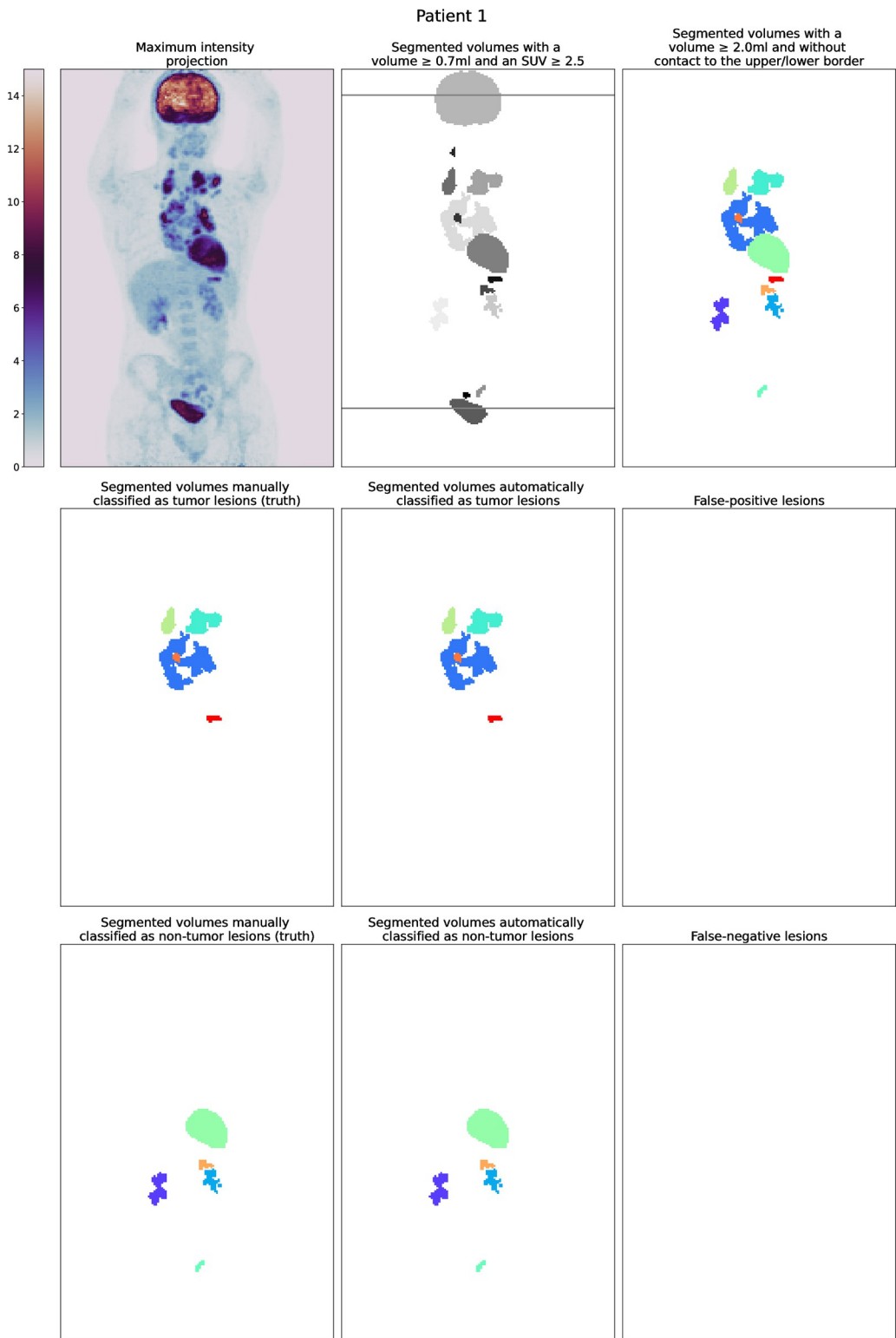

**Fig 2. Patient from the EuroNet-PHL-C1 study with an optimal automatic classification.**

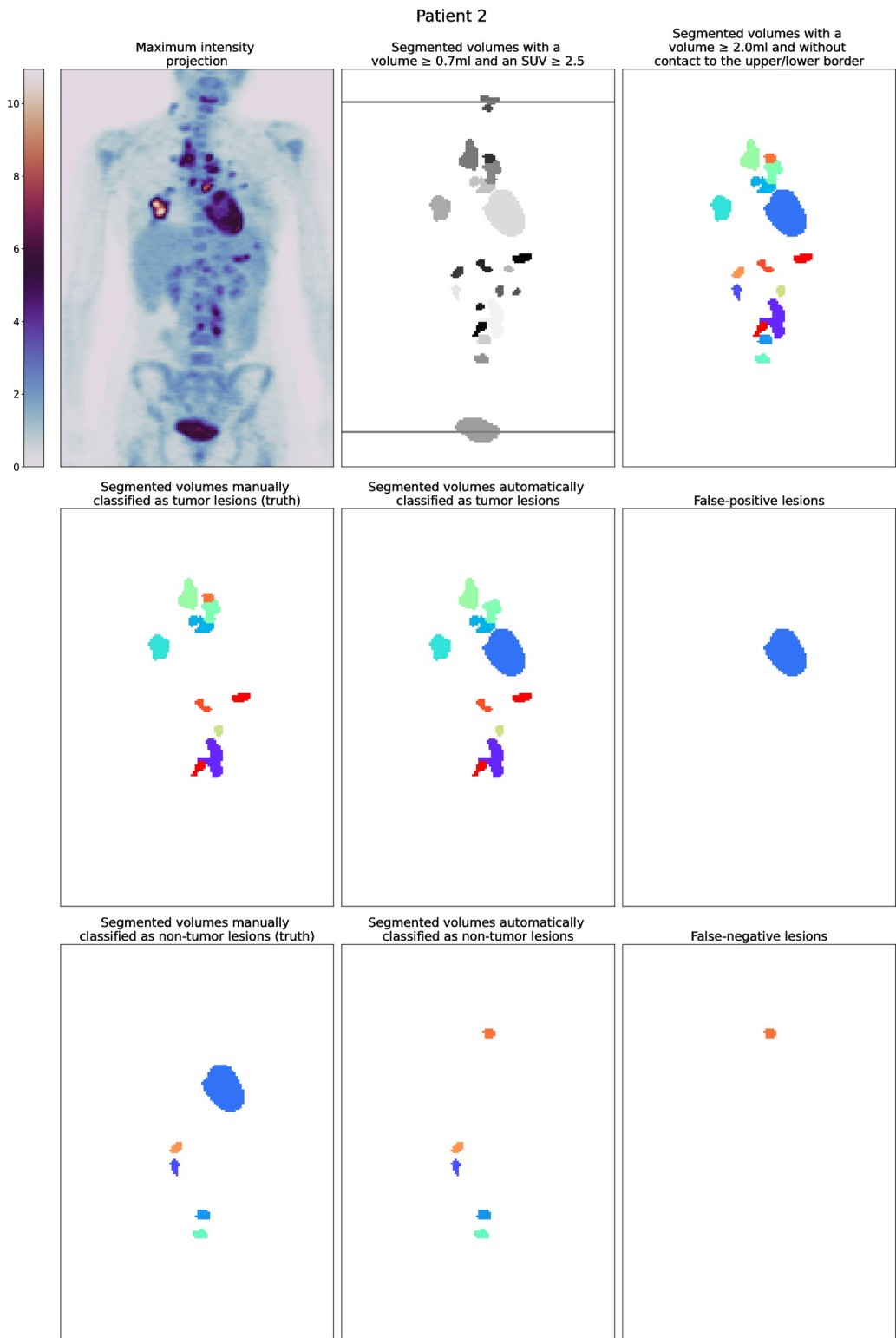

**Fig 3. Patient from the EuroNet-PHL-C1 study with a suboptimal automatic classification and incorrect cropping at the upper border.**

segmentation thresholds were used by other authors: fixed thresholds of an SUV of 2.5 [12, 16], 3.0 [23] or 4.0 [26], or adaptive thresholds of 41%/42% of the SUVmax [11, 16, 26, 27] or background-related thresholds [12, 16, 21].

Second, a watershed algorithm was chosen for segmentation. Watershed segmentation is frequently used for tumor segmentation in PET imaging [20, 23].

Third, the z-axis of the PET scans was cropped for standardization of the scan length. Cropping was important in our multicenter approach since different PET acquisition protocols were used (e.g. skull base to mid-thigh, whole-body). As a result of automatic cropping, only very few lesions got lost.

Fourth, volumes below a size of 2.0 ml were not considered for classification. This approach was also applied by other authors and reduced the number of artifact-related lesions, prone to misclassification [12, 21]. Moreover, a lower threshold on volume size was also used for staging assessment in lymphoma trials [18, 28].

Fifth, a random forest was applied for classification of the segmented volumes. We used several hundred of trees and 31 features for the random forest. Feature standardization was achieved by transformation of the computed features to a standardized size and scanner resolution avoiding inaccuracies introduced by resampling.

Sixth, all tissue types with an increased uptake were included in our evaluation to address the situation in the daily routine. Overall, 14 different categories were used for classification. Most published studies only differentiated between tumor and non-tumor lesions [12, 27], while Hsu et al. [23] used a total of six classification categories.

## Limitations

The main reason for incorrect automatic cropping was the absence of the brain on the PET scan. Our cropping algorithm should be improved regarding this special situation.

Composed volumes consisting of more than one tissue category are a major issue for automatic classification [23, 26]. In our study, 17/21 T+NT volumes and 5/6 NT+NT volumes were classified as tumor lesions. Thus, for our algorithm, composed volumes appeared to resemble tumor rather than non-tumor tissue, regardless of whether they contained tumor or not. Although T+NT volumes were classified as tumor these volumes did not exclusively represent tumor tissue. Thus, the assessment of tumor features could be skewed by the non-tumor parts. Automatic splitting of composed volumes into the individual tissues is not yet possible [26] but might be an option in the future. Manual splitting is a solution for individual cases but not for the assessment of large datasets. A potential approach for automatic splitting might be the organ segmentation in morphological imaging and the transfer of the segmented volume to the PET scan.

## Comparison with other algorithms

A comparison of our results with published data is complicated by differences in methodology, tumor spread in included patients and inclusion of a varying number of non-tumor categories.

A similar methodology for lymphoma classification was published by Hsu et al. [23] using also PET data only, a watershed algorithm and a random forest. The authors identified tumor lesions with sensitivity of 84% and specificity of 92%. Compared to Hsu et al., we trained our algorithm on PET scans from multiple centers, used an automatic cropping algorithm for scan length normalization, included all tissue types with increased uptake in our classification and had more tumor lesions per scan (four vs. one). Interestingly, in our more complex setting we could confirm the very good performance of this methodology.

Several authors used convolutional neural networks (CNN) for automatic segmentation [11, 21] and/or classification [12, 27] of lymphoma lesions. Capobianco et al. [12] and Sibille et al. [27] presented a sensitivity and specificity of 80% and 88%, and of 75% and 96% for lymphoma classification. Both CNN studies used information from PET and corresponding morphological imaging. It should be noted that our algorithm achieved comparable results while only PET imaging and a significantly lower number of manually classified scans was used.

## Conclusion

Our algorithm, trained on a small number of scans and on PET information only, showed a good performance and is suitable for automatic lymphoma classification. Compared to a CNN, a lower number of data is required for sufficient training of our algorithm.

The future method of choice for automatic lymphoma classification is still unclear. Different approaches should be developed and tested to find the optimal solution.

## Supporting information

**S1 Fig. Automatic cropping algorithm.** The aim of heuristic cropping was to detect the brain and the urinary bladder as the cranial and caudal reference points. In most cases both show a distinct physiological uptake and therefore are a viable targets. In a first step the 3D SUV image (a) was projected onto the xz-plane (b), cropped to exclude extremities (c), and further projected onto the z-axis to create a function of SUV maxima from cranial to caudal (d). In a seconded step we smoothed the function and applied a peak detection to extract the first and last significant peak (e). Those correlated in most cases with brain and bladder and served as boundaries for cropping.
(TIF)

**S1 File. Explanation of the nested cross validation.**
(DOCX)

## Author Contributions

**Conceptualization:** Thomas W Georgi, Axel Zieschank, Lars Kurch, Osama Sabri, Dieter Körholz, Christine Mauz-Körholz, Regine Kluge, Stefan Posch.

**Data curation:** Thomas W Georgi, Axel Zieschank, Kevin Kornrumpf, Lars Kurch.

**Formal analysis:** Thomas W Georgi, Axel Zieschank, Kevin Kornrumpf, Stefan Posch.

**Funding acquisition:** Thomas W Georgi, Regine Kluge.

**Investigation:** Thomas W Georgi, Axel Zieschank, Kevin Kornrumpf, Regine Kluge, Stefan Posch.

**Methodology:** Thomas W Georgi, Stefan Posch.

**Project administration:** Dieter Körholz, Regine Kluge, Stefan Posch.

**Resources:** Stefan Posch.

**Software:** Stefan Posch.

**Supervision:** Osama Sabri, Christine Mauz-Körholz, Regine Kluge, Stefan Posch.

**Validation:** Thomas W Georgi, Axel Zieschank, Kevin Kornrumpf, Lars Kurch, Osama Sabri, Dieter Körholz, Christine Mauz-Körholz, Regine Kluge, Stefan Posch.

**Visualization:** Thomas W Georgi, Kevin Kornrumpf, Stefan Posch.

**Writing – original draft:** Thomas W Georgi.

**Writing – review & editing:** Thomas W Georgi, Axel Zieschank, Kevin Kornrumpf, Lars Kurch, Osama Sabri, Dieter Körholz, Christine Mauz-Körholz, Regine Kluge, Stefan Posch.

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
