## [Decision Letter · Decision Letter 0]

8 Mar 2022

PONE-D-22-03090Automatic classification of lymphoma lesions in FDG-PET – Differentiation between tumor and non-tumor uptakePLOS ONE

Dear Dr. Georgi,

Thank you for submitting your manuscript to PLOS ONE. After careful consideration, we feel that it has merit but does not fully meet PLOS ONE’s publication criteria as it currently stands. Therefore, we invite you to submit a revised version of the manuscript that addresses the points raised during the review process.

We look forward to receiving your revised manuscript.

Kind regards,

Domenico Albano

Academic Editor

PLOS ONE

Journal Requirements:

"Supported by the Mitteldeutsche Kinderkrebsforschung."

"The study was supported by the foundation "Mitteldeutsche Kinderkrebsforschung" ("Children's cancer research of central germany")

https://www.kinderkrebsforschung.net

Please change the paper according to reviewers suggestions.

Reviewers' comments:

Reviewer's Responses to Questions

**Comments to the Author**

1. Is the manuscript technically sound, and do the data support the conclusions?

Reviewer #1: Partly

Reviewer #2: Yes

2. Has the statistical analysis been performed appropriately and rigorously? 

Reviewer #1: Yes

Reviewer #2: Yes

3. Have the authors made all data underlying the findings in their manuscript fully available?

Reviewer #1: Yes

Reviewer #2: Yes

4. Is the manuscript presented in an intelligible fashion and written in standard English?

Reviewer #1: Yes

Reviewer #2: Yes

5. Review Comments to the Author

Reviewer #1: #The article is interesting; the authors present an automated algorithm aimed for classification of FDG- uptake in patients with aggressive lymphomas.

#Why did you only include patients with Hodjkin lymphoma? It seems interesting to add patients with other types of lymphoma mainly DLBCL.

#Sensitivity of the algorithm is too low (20% false negative). This is comparable to other algorithms, but hampers the ability to use the algorithm in a real life clinical setting. Please comment on this matter.

#Specifically, low specificity for Gastrointestinal disease might be even more problematic with other types of aggressive lymphoma that tend to involve the GI tract more commonly than Hodjkin lymphoma (eg. DLBCL).

Reviewer #2: Dear authors,

this is a really interesting work about an "hot topic" of future nuclear medicine.

Even if the quality of the study is high, I have some small requests:

1. the articles clearly states that different PET scanners of different centers were used to performe the study. In this setting, more information about the scanner used (PET or PET/CT, analogic or digital scanner, reconstruction algorithm..) are required, better if presented in a table.

2. the "Automatic Classification" section even if really precise is hard to understand and to follow.

3. informations about tumor localization should be given: were all of them nodal localization or other organs were involved?

6. PLOS authors have the option to publish the peer review history of their article (what does this mean?). If published, this will include your full peer review and any attached files.

Reviewer #1: No

Reviewer #2: No

---

## [Author Response · Author response to Decision Letter 0]

21 Mar 2022

Reviewer 1

Why did you only include patients with Hodgkin lymphoma? It seems interesting to add patients with other types of lymphoma mainly DLBCL.

You are right, the classification of patients with non-Hodgkin lymphoma would also be interesting and should be subject of a further study. It would be easily possible to use our method to train different classifiers in a group of DLBCL patients.

Our focus was to develop a classification algorithm for Hodgkin lymphoma patients. Our group is responsible for reference reading of PET scans in the EuroNet - pediatric Hodgkin lymphoma trials. Therefore, our aim is a further improvement of the prognostic information which can be derived from PET. Our next step will be the multifactorial search for prognostic factors for relapse in pretreatment PET of Hodgkin lymphoma patients. The presented automatic classification algorithm is a precondition for this study.

Sensitivity of the algorithm is too low (20% false negative). This is comparable to other algorithms, but hampers the ability to use the algorithm in a real life clinical setting. Please comment on this matter.

It is correct, a sensitivity of 83% probably hampers the use of our classifier in a real life setting, if the algorithm is used alone. However, our automatic algorithm could support the diagnostic work of physicians. Additional information from an automatic classifier will most likely improve the accuracy of the physicians report. Given the complex physiological FDG pattern and the variable occurrence of lymphoma involvement, our results seem to be quite solid, especially since only PET data was used in our study.

Most frequently, false negative lesions occurred in the skeleton and the head-and-neck area. The differentiation between malignant lymph nodes, inflammatory lymph nodes and accessory salivary glands in the neck can be sophisticated, even for experienced physicians. Likewise, the differentiation between physiological increased skeletal uptake and focal skeletal lesions may be also challenging for physicians. In an interreader study of five experts, discrepancies in reporting occurred predominantly in the same areas (Kluge et al. Inter-Reader Reliability of Early FDG-PET/CT Response Assessment Using the Deauville Scale after 2 Cycles of Intensive Chemotherapy (OEPA) in Hodgkin's Lymphoma. PLoS One. 2016;11:e0149072). An additional automatic analysis could help to harmonize the reporting. 

Specifically, low specificity for gastrointestinal disease might be even more problematic with other types of aggressive lymphoma that tend to involve the GI tract more commonly than Hodgkin lymphoma (eg. DLBCL).

The correct classification of physiological gastrointestinal uptake was an issue in our study. However, none of our study patients had lymphoma involvement of the gastrointestinal tract. Only three patients had mesenterial lymph node lesions. Misclassification occurred between patchy intestinal uptake and abdominal lymph node involvement.

In lymphoma types with a tendency of GI involvement, GI lesions would occur frequently and often with large tumor manifestations in this region. An automatic classifier would probably learn how GI lesions look like and how to discriminate them better from non-malignant GI uptake. Thus, the algorithm performance for the classification of GI-lesions would most likely increase significantly.

Reviewer 2

The articles clearly states that different PET scanners of different centers were used to performe the study. In this setting, more information about the scanner used (PET or PET/CT, analogic or digital scanner, reconstruction algorithm..) are required, better if presented in a table.

We agree, it would be interesting to add an overview of the scanners used. Unfortunately, we are not able to present the data since patient data of all C1 patients were completely anonymized first, for data safety reasons. Thereafter, the 60 patients analyzed in our study were drawn randomly out of all C1 patients. Thus, it is not possible anymore to access and present the scanner data.

The "Automatic Classification" section even if really precise is hard to understand and to follow.

Thank you for this comment. We revised the “Automatic classification” section in the methods part. Now, a simplified explanation of the used method is given here. A detailed description can be found in the supplementary material.

Informations about tumor localization should be given: were all of them nodal localization or other organs were involved?

We added the following information in the “Manual classification” section in the result part:

“Tumor lesions occurred in lymph nodes above and below the diaphragm in 59 and 17 patients, respectively. Extra-nodal lesions occurred in the lung, the skeleton, the liver and the spleen in eight, five, one and ten patients, respectively.”

---

## [Decision Letter · Decision Letter 1]

6 Apr 2022

Automatic classification of lymphoma lesions in FDG-PET – Differentiation between tumor and non-tumor uptake

PONE-D-22-03090R1

Dear Dr. Thomas W Georgi,

We’re pleased to inform you that your manuscript has been judged scientifically suitable for publication and will be formally accepted for publication once it meets all outstanding technical requirements.

Kind regards,

Domenico Albano

Academic Editor

PLOS ONE

Reviewers' comments:

Reviewer's Responses to Questions

**Comments to the Author**

1. If the authors have adequately addressed your comments raised in a previous round of review and you feel that this manuscript is now acceptable for publication, you may indicate that here to bypass the “Comments to the Author” section, enter your conflict of interest statement in the “Confidential to Editor” section, and submit your "Accept" recommendation.

Reviewer #1: All comments have been addressed

Reviewer #2: All comments have been addressed

2. Is the manuscript technically sound, and do the data support the conclusions?

Reviewer #1: Yes

Reviewer #2: Yes

3. Has the statistical analysis been performed appropriately and rigorously? 

Reviewer #1: Yes

Reviewer #2: Yes

4. Have the authors made all data underlying the findings in their manuscript fully available?

Reviewer #1: Yes

Reviewer #2: Yes

5. Is the manuscript presented in an intelligible fashion and written in standard English?

Reviewer #1: Yes

Reviewer #2: Yes

6. Review Comments to the Author

Reviewer #1: The manuscript is solid and intresting. The authors adressed my comments and explained why some of them are not in the scope of the cureent study.

Reviewer #2: (No Response)

7. PLOS authors have the option to publish the peer review history of their article (what does this mean?). If published, this will include your full peer review and any attached files.

Reviewer #1: No

Reviewer #2: No

---

## [Editor Report · Acceptance letter]

8 Apr 2022

PONE-D-22-03090R1 

Automatic classification of lymphoma lesions in FDG-PET – Differentiation between tumor and non-tumor uptake 

Dear Dr. Georgi:

I'm pleased to inform you that your manuscript has been deemed suitable for publication in PLOS ONE. Congratulations! Your manuscript is now with our production department. 

Kind regards, 

on behalf of

Dr. Domenico Albano 

Academic Editor

PLOS ONE